# The Nexus between Entrepreneurial Education and Entrepreneurial Self-Competencies: A Social Enterprise Perspective

**Frank Frimpong Opuni** [1,*], **Michael Snowden** [2,*], **Ernest Christian Winful** [3], **Denis Hyams-Ssekasi** [4], **Jamie P. Halsall** [2], **Josiah Nii Adu Quaye** [3], **Emelia Ohene Afriyie** [5], **Elikem Chosniel Ocloo** [1] **and Kofi Opoku-Asante** [3]

[1] Marketing Department, Accra Technical University, Accra P.O. Box GP 561, Ghana
[2] School of Human and Health Sciences, University of Huddersfield, Huddersfield HD1 3DH, UK
[3] Accounting and Finance Department, Accra Technical University, Accra P.O. Box GP 561, Ghana
[4] Institute of Management, University of Bolton, Bolton BL3 5AB, UK
[5] Management and Public Administration, Accra Technical University, Accra P.O. Box GP 561, Ghana
* Correspondence: fofrimpong@atu.edu.gh (F.F.O.); m.a.snowden@hud.ac.uk (M.S.)

**Abstract:** The purpose of the study was to examine the mediation roles of student satisfaction and entrepreneurial self-efficacy in the nexus between entrepreneurial education and entrepreneurial self-competencies within a social enterprise context. The study used a cross-sectional survey design, with a sampled population of 185 business students from three universities (Accra Technical University, Cape Coast Technical University and the University of Ghana) in Ghana. A PLS-SEM approach was used to examine the relationships among the independent–dependent constructs in the study. Entrepreneurial education had positive and significant relationships to student satisfaction and entrepreneurial self-efficacy, but it showed an insignificant relationship to entrepreneurial self-competencies. Student satisfaction was also found to relate positively and significantly to entrepreneurial self-efficacy and entrepreneurial self-competencies. Furthermore, both student satisfaction and entrepreneurial self-efficacy were found to fully mediate the nexus between entrepreneurial education and entrepreneurial self-competencies. The study highlights the crucial roles of student satisfaction and self-efficacy in the implementation of entrepreneurial education in higher education institutions. In a discipline that is characterised by paucity, this study provides a unique and original assessment of the important roles of student satisfaction and student self-confidence in building entrepreneurial competencies among students.

**Keywords:** entrepreneurial education; student satisfaction; entrepreneurial self-efficacy; entrepreneurial self-competency; social enterprise

## 1. Introduction

Unemployment has assumed alarming proportions amongst young people graduating from higher education institutions (HEIs) within the sub-Saharan region of Africa. Ghana is no exception, with an approximately 13.4% unemployment rate as of the end of 2021 (Ghana Statistical Service). Young people account for the majority of the unemployed (32% of youth between 15 and 24 years of age are unemployed) population in Ghana. This situation calls for a paradigm shift in the curriculum of HEIs in order to stem the tide of rising youth unemployment. Social enterprise has been postulated as having great potential to reduce youth unemployment since the model has proven successful in other emerging economies [1]. However, its effective implementation and sustainability are dependent on the effectiveness of entrepreneurship education in HEIs [2] and the definition and classification of social enterprise [3,4].

There is currently an ongoing debate as to whether contemporary entrepreneurial pedagogy in Ghana leads to the acquisition of entrepreneurial skills for graduating youth [5,6]. However, there is a paucity of valid studies examining the effect of entrepreneurship education on educational competencies for nascent and early-stage young graduates in Ghana, especially from a social enterprise perspective. This has resulted in a dearth of research-based output to inform stakeholders on policy formulation and implementation. Moreover, studies within the field typically appear to suffer from a lack of robust analyses and methodological flaws [7]. There is a clear scholarly deficit in terms of studies examining the relationships between entrepreneurship education, self-efficacy, student satisfaction and entrepreneurial competencies from a social enterprise perspective in the sub-Saharan region in general, and in Ghana in particular. Within the extant literature, the effectiveness of entrepreneurial education has been linked to student satisfaction, which could engender entrepreneurial self-efficacy and ultimately influence entrepreneurial competency. However, there is a paucity of studies on the aforementioned constructs within the region, with a special emphasis on Ghana. Moreover, the majority of recent studies in Ghana have focused on entrepreneurial education, skill acquisition and intention [5,8–12]. It is also illuminating that these studies have concentrated on traditional private-sector-led enterprises, with none of them addressing these issues from a social enterprise standpoint.

The current study addresses this scholarly gap and provides information to key social enterprise stakeholders and ecosystem players to inform and generate policies and interventions to help foster sustainable employment in addressing the eighth United Nations sustainable development goal (SDG), which aims to promote inclusive and sustainable economic growth, employment and decent work for all men and women. In addition, findings from the study will help to improve the quality of entrepreneurship education in HEIs and hence address the third SDG, which aims to ensure inclusive and quality education for all and to promote lifelong learning. The study's outcomes will also provide key stakeholders, such as government, HEIs, students, academia and industry, with the requisite information for policy formulation, decision making and implementation regarding social enterprise development in Ghana. Given the foregoing discussion, the key research questions in the current study are: (1) What is the effect of entrepreneurship education on entrepreneurial competencies? and (2) To what extent is this relationship mediated by student satisfaction and self-efficacy? Answers to the above questions will help prescribe the pedagogical basis for entrepreneurial education in the form of learning strategy designs, assessment techniques and entrepreneurial training that can help improve the entrepreneurial competencies of students within a social enterprise context. Furthermore, this study serves as a response to recent calls by Bolzani and Luppi [13] and Lv et al. [14] for further empirical studies on the impact of entrepreneurial education on students' entrepreneurial competencies in higher education institutions.

Based on the foregoing premise, this study investigated the prevalence of eight pieces of empirical evidence in social enterprise education in the higher education sector in Ghana. They are as follows: (1) Entrepreneurial education is positively related to entrepreneurial self-efficacy; (2) Entrepreneurial education improves student satisfaction; (3) Student satisfaction improves entrepreneurial self-efficacy; (4) Entrepreneurial self-efficacy improves entrepreneurial self-competencies; (5) Student satisfaction improves entrepreneurial self-competencies; (6) Entrepreneurial self-efficacy improves entrepreneurial self-competencies; (7) Student satisfaction mediates the relationship between entrepreneurial education and entrepreneurial self-competencies; and (8) Entrepreneurial self-competency mediates the relationship between entrepreneurial education and entrepreneurial self-competencies. The study, therefore, contributes to the academic debate on the effect of entrepreneurial education on entrepreneurial self-efficacy and entrepreneurial self-competencies from a developing economy perspective. It also establishes empirical evidence on the relevance of student satisfaction in the delivery of entrepreneurial education in the higher education sector in Ghana. The remainder of the study is organised

under the following headings: Materials and Methods, Methodology, Results, Discussion, Theoretical Implications, Managerial and Practical Applications, and Limitations and Future Studies.

## 2. Materials and Methods

### 2.1. Theoretical Framework and Hypotheses Development

The social cognition theory (SCT) of Albert Bandura [15] postulates that the environment influences behaviour, and behaviour, in turn, affects the environment. This kind of mutual interplay between the environment and behaviour is known in the literature as reciprocal determinism. Given this premise, it can be suggested that the entrepreneurial cognition, affect and behaviour of students can be influenced by the educational environment in which they find themselves. However, it can also be suggested that the entrepreneurial behaviours exhibited by students within their communities can positively influence societal resilience and community fulfilment.

The current study seeks to apply the concepts found in SCT in the development of the conceptual framework. The underlying assumption is that each student's interactions with entrepreneurial learning approaches, current pedagogy, and heutagogy, as well as with entrepreneurial mentors and coaches (the environment), can reinforce entrepreneurial cognition and behaviour. In terms of the current study, it is posited that student interaction with the entrepreneurial environment enhances cognition, engenders satisfaction and further leads to desired entrepreneurial behaviours, which include entrepreneurial self-efficacy and competencies. The SCT is therefore employed to generate the conceptual framework for the current study.

The hypothesised relationships for the current study are presented in the framework (Figure 1). The main independent or exogenous variable is entrepreneurial education, which is deemed to have a direct and positive relationship to the main dependent variable or endogenous variable, i.e., entrepreneurial self-competency. This relationship is deemed to be non-linear since the link between the two constructs is deemed to be mediated in a parallel fashion by student satisfaction and entrepreneurial self-efficacy. This framework assumes a complex network of independent and dependent variables so that a structural equation modelling approach is adopted to analyse the constructs simultaneously [16].

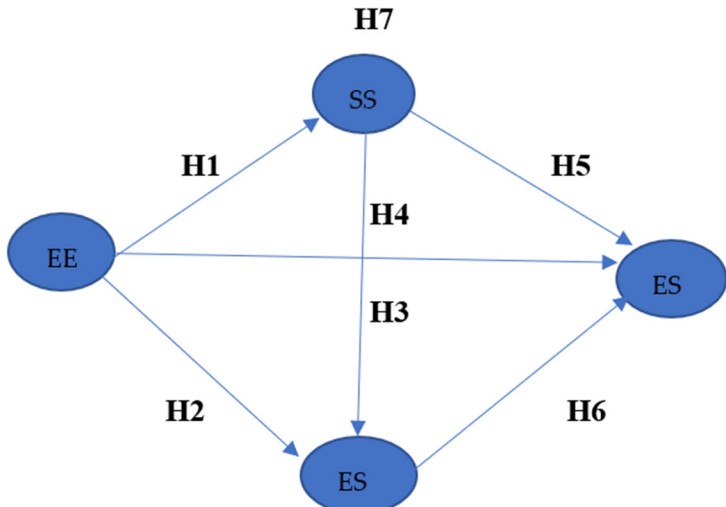

**Figure 1.** Research model.

### 2.2. Entrepreneurial Education and Student Satisfaction

According to Liu et al. [17], entrepreneurial education refers to activities that aim to develop and improve the business mindset, aspirations and confidence required for the

student to adequately design their own business plan and start and own an enterprise. Maritz and Brown [18] suggested that entrepreneurship education enlightens students about the importance of embracing challenges and having a penchant for innovation.

Satisfaction is defined as the feeling expressed by a person pleased by some service that meets or exceeds their expectations [19]. Relatedly, customer satisfaction is generally seen as an appraisal of the services being provided by the business in terms of the user's experience at the point of delivery of that particular service [20]. Since students are customers of higher education institutions, student satisfaction can be defined as the feelings expressed by students in terms of the services meeting or exceeding their expectations. Student satisfaction has also been linked with the quality of service provided by the university. From a marketing perspective, the student is seen as a customer obtaining some form of service from the educational organisation. Therefore, the provision of quality entrepreneurial education can be conceptualised from a service delivery perspective. If this is the case, then the quality of entrepreneurial education can be associated with student satisfaction, just as service quality provision has been associated with customer satisfaction in the marketing literature. With reference to Parasuraman's five-dimensional SERVQUAL model, reliability is seen as the ability to provide the promised service reliably and correctly; responsiveness is described as the willingness to assist customers and meet their needs; tangibility refers to the physical appearance of service providers, physical assets and equipment; guarantee is seen as the ability of the company to assure customers of the promised service and inspire confidence; and empathy is the customised service that is uniquely offered to the client. Hasan et al. [21] posit that there is a positive correlation between these five antecedents of service quality and student satisfaction. Brown and Mazzarol [22] and Dericks et al. [23] suggested that within the arena of higher education, a good number of empirical studies offer support for the existence of a positive nexus between service quality and student satisfaction.

Within the literature, it has been established in a good number of studies that entrepreneurial education has a direct relationship with student satisfaction. Since student satisfaction is a tool to measure the quality of education [24], it can be suggested that quality entrepreneurial education can have a positive impact on student satisfaction. In other words, the content and methods of an entrepreneurial programme are key in fostering student satisfaction [25]. The competence level of lecturers, teaching methods, learning facilities and student support services are key elements that can engender student satisfaction in the context of entrepreneurial education. These recent studies indicate that an improvement in entrepreneurial education leads to increased student satisfaction levels [25,26]. However, results from the studies of Long et al. [27] in Cambodia pointed to an insignificant relationship between entrepreneurial education and student satisfaction. Huang et al. [25], for instance, found that the relationship between entrepreneurial education and student satisfaction was mediated by entrepreneurial practice. There is, as yet, still a paucity of studies on the satisfaction derived from entrepreneurial education in HEIs [25], especially within a developing economy context. Given the foregoing discussions on the potential relationship between entrepreneurial education and student satisfaction, we posit that:

**H₁:** *Entrepreneurial education has a positive and significant influence on student satisfaction.*

*2.3. Entrepreneurial Education and Entrepreneurial Self-Efficacy*

According to Nazmi [28], the major objective of entrepreneurship education is to train students to become self-employed or highly productive in the workplace. This indirectly connotes that in order for students to become self-dependent, they need a good measure of self-belief or self-confidence. Bandura and Adams [29] define self-efficacy as a person's belief in their ability to perform a given task. Self-efficacy can also be conceptualised as a self-assessment of one's ability to successfully finish a task along with the confidence that one possesses the skills needed to complete the job [30]. By extension,

entrepreneurial self-efficacy can be defined as the extent of belief in one's own internal capacity to accomplish entrepreneurial tasks or behaviours in a given situation [31]. In other words, entrepreneurial self-efficacy is an expression of the individual's self-confidence in their ability to perform a certain task connected to entrepreneurship [17].

There is a body of evidence in the literature that confirms a positive association between entrepreneurial education and entrepreneurial self-efficacy [32–37]. In a recent study by Zhang et al. [38] sampling 910 students across three universities in China, the authors, using a stepwise regression method, reported that students' positive attitudes towards entrepreneurial education exerted a positive and significant influence on entrepreneurial self-efficacy. Relatedly, and from an emerging economy perspective, the results of research conducted by Memon et al. [39] on university students in Pakistan found that the acquisition of enterprise development knowledge from an entrepreneurship education programme had a significant influence on their entrepreneurial self-efficacy. Abaho et al. [19], exploring 522 final-year students from selected universities in Uganda, reported that entrepreneurial education, in the form of exposure to lecturers with business experience, instilled entrepreneurial self-efficacy among them. Given the foregoing discussions, we hereby posit that:

**H2:** *Entrepreneurial education has a positive and significant influence on entrepreneurial self-efficacy.*

### 2.4. Entrepreneurial Education and Entrepreneurial Self-Competencies

Entrepreneurial competencies refer to a mix of entrepreneurial abilities, which comprise initiatives or motivations, a set of knowledge, a field of speciality, attitudes or insights, and personal qualities or characteristics, that can facilitate practical business solutions [40]. In other words, entrepreneurial self-competencies consist of innovativeness, proactiveness and taking initiative to provide business solutions [41]. Within the extant literature, the concept of entrepreneurial self-competencies has been defined to include these key dimensions: functional competencies, personal competencies, technological competencies, interpersonal competencies, environmental competencies and ethical competencies.

According to Hagg and Gabrielson [42], there is empirical evidence to show that entrepreneurial education directly influences entrepreneurial self-competencies. In an exploratory study in Malaysia by Bagheri and Lope Pihie [43], the authors reported that entrepreneurial education had a profound and significant effect on students' business and entrepreneurial self-competencies. In addition, other previous studies point to a positive association between the variables [32,44–46]. Given the foregoing findings, the following directional relationship is suggested:

**H3:** *Entrepreneurial education has a positive and significant influence on entrepreneurial self-competencies.*

### 2.5. Student Satisfaction and Entrepreneurial Self-Efficacy

According to Saeed et al. [47], the provision of satisfying experiences in the entrepreneurship learning environment can significantly increase entrepreneurial self-efficacy among students. This can be achieved through the promotion of learning by doing such things as engaging in fulfilling apprenticeships, drawing business plans, listening to accomplished entrepreneurs and undergoing business simulations. Ahmed et al. [48] reported in their studies in Pakistan (an emerging economy) that student satisfaction had a direct influence on entrepreneurial self-efficacy. The study found that satisfied students had higher levels of entrepreneurial self-efficacy, but dissatisfied students had lower levels of entrepreneurial self-efficacy. Thus, students' satisfaction level, derived from the level of service quality of an entrepreneurship education programme, had the tendency to positively influence students' entrepreneurial self-efficacy and, further, inform their

readiness to start their own businesses. In a similar study in Taiwan by Yen and Lin [49], the authors found that business students' flow experience led to superior satisfaction, which, in turn, increased entrepreneurial self-efficacy. Combing through the literature, there is an apparently low level of scholarly attention paid to entrepreneurial self-efficacy as an outcome measure of student satisfaction. Based on the foregoing, we posit that:

**H4:** *Student satisfaction has a positive and significant relationship with entrepreneurial self-efficacy.*

### 2.6. Student Satisfaction and Entrepreneurial Self-Competency

There is some evidence to suggest that students who are satisfied with entrepreneurship learning develop the basic entrepreneurial self-competencies necessary to kickstart their own businesses. A study conducted by Edeling and Pilz [50] in the vocational education landscape across Germany, Poland and Italy found that student satisfaction had a positive and significant association with self-competencies. Plumly et al. [51] established that students who gained satisfaction through business experience developed key entrepreneurial competencies, such as effective communication, good negotiation skills, working in teams, meeting legal requirements and exercising creativity and innovation. In a study involving 710 students in Siberia, Ustyuzhina et al. [52] reported that about 78% of them expressed a lack of readiness to start their own businesses due to the fact that they lacked key competencies. The respondents attributed the lack of competencies to their dissatisfaction with the entrepreneurship education process. Among other causes, their apparent dissatisfaction was attributed to teacher incompetence, the low number of business visits and trips, the lack of experience developing business plans and the lack of internships and work with entrepreneurs. Based on the foregoing discussions, we posit that:

**H5:** *Student satisfaction has a positive and significant relationship with entrepreneurial self-competencies.*

### 2.7. Entrepreneurial Self-Efficacy and Entrepreneurial Self-Competency

The literature suggests that self-efficacy can influence self-competency [53]. The reasoning behind this relationship is that students who have a good sense of self-belief and achievement strive to develop the key competencies that can help them achieve their goals in their chosen fields of endeavour. Conversely, individuals who lack confidence in themselves may not avail themselves of opportunities or have belief in their abilities to acquire certain skills and competencies. Findings from previous studies [54–57] suggest a positive association between entrepreneurial self-efficacy and self-competency. Students, as potential social entrepreneurs, need to demonstrate self-efficacy that can help them develop social self-competencies. In this regard, Urban [58] suggests that entrepreneurial self-efficacy is vital in developing social self-competencies, which, in turn, are crucial in generating socially desirable outcomes within the context of social enterprise. Given the foregoing discussions, we hereby posit that:

**H6:** *Entrepreneurial self-efficacy has a positive and significant relationship with entrepreneurial self-competencies.*

### 2.8. The Mediation Roles of Student Satisfaction and Entrepreneurial Self-Efficacy

Satisfaction is seen as a key outcome in the provision of entrepreneurial education. Razinkina et al. [59] posit that satisfaction with the curriculum is a measure of the quality of service provided by the institution. There is anecdotal evidence to suggest that satisfaction can enhance the relationship between entrepreneurial education, entrepreneurial efficacy and entrepreneurial competencies. In other words, the literature suggests that student satisfaction tends to increase the effect of entrepreneurial education on students' en-

trepreneurial efficacy. Ustyuzhina et al. [52] and Bramante [60] found evidence that entrepreneurial self-efficacy mediates the relationship between entrepreneurial education and entrepreneurial self-competency. Therefore, the following hypotheses are posited:

**H7:** *Student satisfaction mediates the relationship between entrepreneurial education and entrepreneurial self-efficacy.*

**H8:** *Entrepreneurial self-efficacy mediates the relationship between entrepreneurial education and entrepreneurial self-competencies.*

### 3. Methodology

*3.1. Sample and Data Collection*

The study was a cross-sectional study targeted at students from three higher education institutions in Ghana: Accra Technical University (ATU), Cape Coast Technical University (CCTU) and the University of Ghana (UG), City Campus (Table 1). The study adopted the three-level stepwise sampling technique of a general, targeted and accessible sampling strategy, as described by Asiamah et al. [61]. Thus, we first applied a simple random method to select 1000 regular final-year marketing students from these universities (293 from ATU, 284 from CCTU and 423 from UG) to form our population. This approach was adopted because the researchers did not know the actual population sizes of the three universities. These final-year marketing students also met certain inclusion criteria: (1) a full-time student; (2) ability to complete questionnaires in English; (3) readiness and willingness to complete the survey. Secondly, by estimation, Yamen's formula [62] was used to obtain a representative sample of 286. Thirdly, stratified sampling by proportional allocation to size via simple random sampling was adopted, with 84 students representing ATU, 81 students representing CCTU and 121 representing UG. This process was performed because the researchers wanted to avoid any form of bias and have a true representation of the population. Among the 286 students, 87% (250) agreed to provide their contact information for the dissemination of the questionnaire by the researchers. Of the 250 contacts provided, 82% (205) students were accessible, and 18% (45) students were not reachable (because they had travelled, were indisposed or simply could not be contacted). A questionnaire, designed using a Google form with a link, was sent to the email addresses of the 205 respondents. Out of 205 invited respondents, 90.7% (186) returned the questionnaires; these responses were used for the analysis. This number was higher than the number of samples used by Aruștei [63] and Robson et al. [64] in their studies. Hair et al. [65,66] assert that PLS-SEM requires neither a large sample size nor a specific assumption in terms of the distribution of the data. The authors posit that researchers working with small sample sizes and less theoretical support for their research can apply PLS-SEM to test causal relationships. The remaining 9.3% of the questionnaires (19) were found to be incomplete and consequently excluded from the analysis (see Supplementary Materials).

**Table 1.** Sampling profile of the students.

| Universities | Population (Strata) | Proportions (x/1000) | Representative Sample (x/1000) × 286 |
|---|---|---|---|
| Accra Technical University | 293 | 0.293 | 84 |
| Cape Coast Technical University | 284 | 0.284 | 81 |
| University of Ghana, City Campus | 423 | 0.423 | 121 |
| Total | 1000 | 1 | 286 |

x—the stratum of the universities' business faculty students.

$$\text{Yamen formula (n)} = \frac{N}{1-N(e)^2} \tag{1}$$

where

n is the desired sample size

*N* is the population size

*e* is the margin of error

$$= \frac{1000}{1-1000(0.05)^2} = 285.7 \approx 286$$

### 3.2. Study Design

A quantitative research method was applied, with a cross-sectional survey design, to test the hypotheses formulated. A correlational technique was most appropriate for testing the nature of the relationships between the identified constructs, namely entrepreneurial education, entrepreneurial self-efficacy, student satisfaction and entrepreneurial self-competency. Since the conceptual framework involved a mix of multiple dependent and independent variables, as well as the structural relationships between independent, mediating and dependent variables, the partial least squares approach to structural equation modelling (PLS-SEM) was used to test the nature of the relationships [61]. This allowed for the prediction of the relationships between the variables identified in this study. Secondly, the PLS-SEM approach was adopted due to the fact that the primary data were not normally distributed and the sample size was small [62].

### 3.3. Measurement Scales

Close-ended self-reported questionnaires were used in this study to gather primary data. The questionnaire consisted of two major parts. The first part was for the collection of data on demographic variables, which included gender, age and educational level. The second part of the questionnaire contained the variables that measured the four main variables described in this study. Accordingly, entrepreneurial education, entrepreneurial self-efficacy, student satisfaction and entrepreneurial self-competencies were measured using validated standard scales that were used in previous empirical studies. Entrepreneurial education was measured with an 8-item entrepreneurial education scale adapted from the entrepreneurial education scale described by Puni et al. [11]. The scale recorded a Cronbach's alpha value of 0.919. In measuring student satisfaction, the study adopted a 7-item student satisfaction scale based on the standard scales used by Nadiri et al. [67] and Wilkins et al. [68]. The adopted scale was revalidated and scored a Cronbach's alpha value of 0.930. The seven-dimension entrepreneurial self-efficacy scale used for the study was adopted from Ehrlich [69]. The adopted scale scored a Cronbach Alpha value of 0.922. The entrepreneurial competencies scale adopted for this study was a standard 8-item scale adapted from Wei [70]. This scale recorded a Cronbach's alpha value of 0.955. Therefore, each scale met the minimum convergent validity criteria of a Cronbach's alpha threshold of 0.5 [61], and all factors loaded more than 0.7, or 70%, onto the latent variables, thereby meeting internal consistency and usability requirements [71] (See Figure 2). The standard scales, therefore, met all reliability and validity criteria for model measurement. The scales for the current study used a 5-point descriptive anchor (1-Strongly disagree, 2-Disagree, 3-Somewhat disagree, 4-Agree and 5-Strongly agree) in measuring each variable.

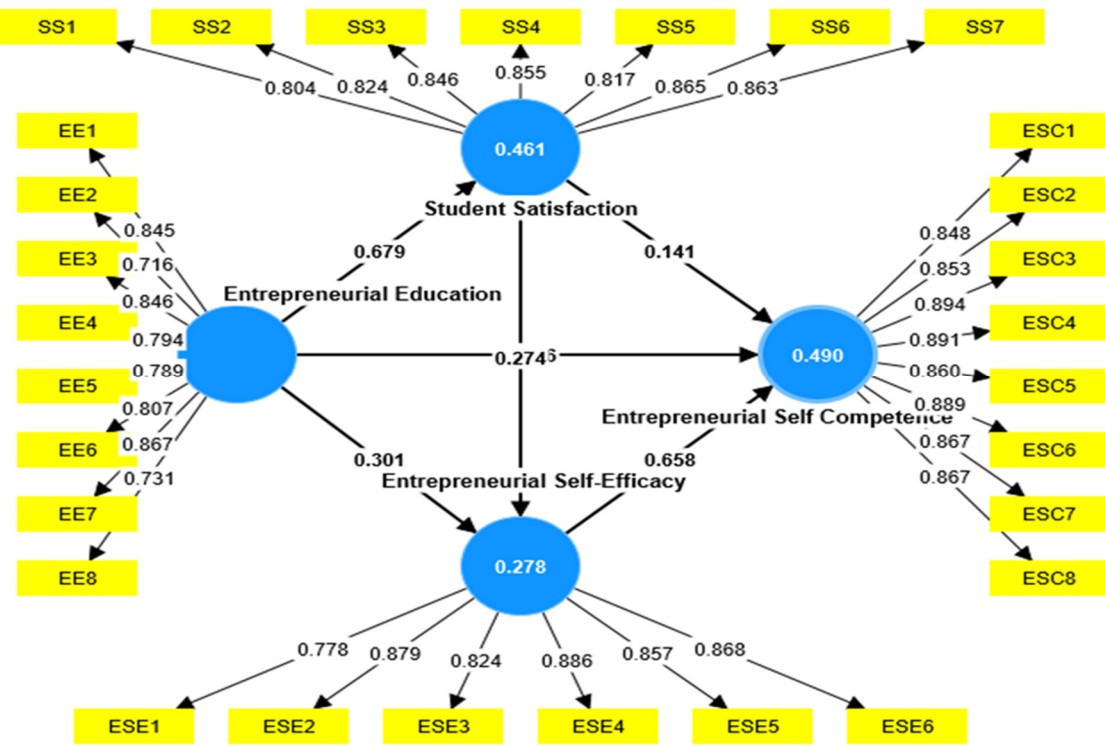

**Figure 2.** Structural model showing path coefficient values.

*3.4. Non-Response and Common Methods Bias*

In the current study, steps were taken to account for non-response and common methods bias. Non-response and common method bias are likely to be associated with cross-sectional studies in which data measuring both independent and dependent variables are gathered from the same sample. First, we split respondents' data into two sets. We then compared the means for the first set to the means for the second set, using the independent samples *t*-test. The outcome for the test was insignificant (i.e., $p > 0.05$), in line with the criteria recommended by Armstrong and Overton [72]. Therefore, non-response bias was not an issue in this study. Next, we employed the full collinearity test method propounded by Kock [73] (2015) to account for common method bias in PLS-SEM. Kock posited that for a model to be free of common method bias, all variance inflation factors (VIFs) in the inner model should be less than or equal to 3.3. Since all the VIFs for all the variables in this study were less than 3.3 (see Table 2), we submit that the model employed for the current study is free of common method bias. Harman's single-factor test was also conducted to test for the occurrence of common methods bias. The first factor was found to account for 44.7% (See Appendix A). Since this value was less than 50%, the model can be said to be free of common method bias [74].

**Table 2.** VIF values for the inner model.

| Construct | EE | ESC | ESE | SS |
|---|---|---|---|---|
| EE | <0.001 | 1.983 | 1.857 | 1.000 |
| ESC | <0.001 | <0.001 | <0.001 | <0.001 |
| ESE | <0.001 | 1.385 | <0.001 | <0.001 |
| SS | <0.001 | 1.961 | 1.857 | <0.001 |

**4. Results**

The PLS-SEM tool was used to analyse data for the current study. Two major steps are identified in the literature: the measurement model and the structural model [61]. The measurement model estimates construct validity and reliability, while the structural

model assesses the nature of the relationships, or the paths, between the key constructs of the study, thereby serving as the test for the hypotheses developed for the study.

*4.1. Measurement Model Assessment*

The measurement model's suitability was assessed according to reliability, convergent validity and discriminant validity criteria. First, we assessed the reliability of the model based on the values of composite reliability and Cronbach's alpha. Hair et al. [65] posit that a measurement model is considered reliable if the Cronbach's alpha and composite reliability figures for all constructs are greater than 0.7. Here, since the values for Cronbach's alpha and composite reliability are greater than 0.7 (see Table 3), the model's reliability is confirmed. Secondly, we assessed the measurement model for convergent validity. Convergent validity is met when the average variance (AVE) extracted for all constructs is greater than 0.5 [72]. Table 3 gives evidence that the AVEs for all four of the constructs in this study are greater than 0.5; hence, convergent validity has been met. Finally, we accounted for the discriminant validity of the measurement model based on the Fornell–Larcker [75] criterion, heterotrait–monotrait (HTMT) criterion and cross-loadings. Fornell–Lacker [75] posits that for discriminant validity, the square root of the AVE for any variable must be greater than the correlation between the said variable and all other variables. Since this condition is met for the model used herein, it can be confirmed that the Fornell–Larcker [75] criterion for discriminant validity is satisfied. The HTMT criterion for discriminant validity requires that HTMT values should be ≤0.90. As shown in Table 4, the HTMT values were all lower than ≤0.90; as such, we can conclude that the respondents understood that the four constructs were distinct. The cross-loadings criterion requires that all items load the highest on their respective constructs. Since Table 5 gives evidence for this requirement, discriminant validity is deemed to be met. Based on the foregoing adequacy measures, our model meets the required psychometric properties of a measurement model.

**Table 3.** Measurement model results.

| Constructs | Cronbach's Alpha | CR | AVE |
|---|---|---|---|
| Entrepreneurial Education | 0.919 | 0.935 | 0.642 |
| Entrepreneurial Self-Competency | 0.955 | 0.962 | 0.759 |
| Entrepreneurial Self-Efficacy | 0.922 | 0.939 | 0.722 |
| Student Satisfaction | 0.930 | 0.943 | 0.704 |

**Table 4.** HTMT values.

| Constructs | EE | ESE | ESC | SS |
|---|---|---|---|---|
| Entrepreneurial Education | <0.001 | <0.001 | <0.001 | <0.001 |
| Entrepreneurial Self-Competency | 0.374 | <0.001 | <0.001 | <0.001 |
| Entrepreneurial Self-Efficacy | 0.529 | 0.736 | <0.001 | <0.001 |
| Student Satisfaction | 0.730 | 0.434 | 0.514 | <0.001 |

**Table 5.** Cross-loadings.

| Constructs | EE | ESC | ESE | SS |
|---|---|---|---|---|
| EE1 | 0.845 | 0.310 | 0.370 | 0.596 |
| EE2 | 0.716 | 0.276 | 0.357 | 0.448 |
| EE3 | 0.846 | 0.301 | 0.375 | 0.559 |
| EE4 | 0.794 | 0.286 | 0.402 | 0.573 |
| EE5 | 0.789 | 0.263 | 0.378 | 0.528 |
| EE6 | 0.807 | 0.331 | 0.474 | 0.560 |
| EE7 | 0.867 | 0.236 | 0.404 | 0.565 |

| | | | | |
|------|-------|-------|-------|-------|
| EE8 | 0.731 | 0.232 | 0.353 | 0.508 |
| ESC1 | 0.383 | 0.848 | 0.598 | 0.400 |
| ESC2 | 0.317 | 0.853 | 0.615 | 0.368 |
| ESC3 | 0.285 | 0.894 | 0.603 | 0.374 |
| ESC4 | 0.328 | 0.891 | 0.589 | 0.366 |
| ESC5 | 0.294 | 0.860 | 0.552 | 0.308 |
| ESC6 | 0.287 | 0.889 | 0.593 | 0.345 |
| ESC7 | 0.269 | 0.867 | 0.615 | 0.368 |
| ESC8 | 0.282 | 0.867 | 0.654 | 0.332 |
| ESE1 | 0.420 | 0.526 | 0.778 | 0.362 |
| ESE2 | 0.438 | 0.619 | 0.879 | 0.446 |
| ESE3 | 0.360 | 0.575 | 0.824 | 0.385 |
| ESE4 | 0.443 | 0.587 | 0.886 | 0.365 |
| ESE5 | 0.409 | 0.621 | 0.857 | 0.437 |
| ESE6 | 0.415 | 0.597 | 0.868 | 0.439 |
| SS1 | 0.644 | 0.367 | 0.414 | 0.804 |
| SS2 | 0.536 | 0.293 | 0.340 | 0.824 |
| SS3 | 0.564 | 0.407 | 0.412 | 0.846 |
| SS4 | 0.537 | 0.323 | 0.403 | 0.855 |
| SS5 | 0.522 | 0.355 | 0.385 | 0.817 |
| SS6 | 0.578 | 0.310 | 0.409 | 0.865 |
| SS7 | 0.593 | 0.349 | 0.440 | 0.863 |

*4.2. Structural Model Assessment*

4.2.1. Path Analysis

Having verified the suitability of the measurement model (Figure 2), the next step was to assess the structural model. The model was found to meet the goodness of fit criteria for PLS-SEM; the normed fit index (NFI) and standardised root mean square residual (SRMR) figures of 0.841 and 0.05, respectively (see Table 6), fell within the acceptable ranges posited by Henseler et al. [76,77] and Bentler and Bonett [78]. Henseler et al. suggest that the NFI value should be less than 1, whilst Bentler and Bonnet suggest that the SRMR figure should be closer to 1.

**Table 6.** Model fit indices.

| Criteria | Saturated Model | Estimated Model |
|------------|-----------------|-----------------|
| SRMR | 0.050 | 0.050 |
| d ULS | 1.080 | 1.080 |
| d G | 0.777 | 0.777 |
| Chi-square | 777.855 | 777.855 |
| NFI | 0.841 | 0.841 |

Secondly, Hair et al. [66] (2014) Bentler and Bonett[80] (1980) propose that for goodness of fit to be established in PLS-SEM, the coefficient of R2 for the dependent construct should be well predicted by the exogenous variables in the path model. As indicated in Figure 2, the R2 value for entrepreneurial self-competency (0.490) shows that 49.0% of the total change in the endogenous construct of entrepreneurial self-competency can be attributed to changes in exogenous constructs, i.e., entrepreneurial education, student satisfaction and entrepreneurial-self-efficacy. Secondly, the R2 value for student satisfaction (0.461) signifies that 46.1% of the total variation in the endogenous construct student satisfaction was due to changes in the exogenous variable entrepreneurial education. Finally, the R2 value of the endogenous variable entrepreneurial self-efficacy (0.278) shows that

27.8% of the total change in the endogenous construct was due to changes in the exogenous variables student satisfaction and entrepreneurial education. Giao and Vuong [79] suggest that changes in endogenous variables emanating from changes in the exogenous variables can be classified according to R2 values of 0.02 (weak), 0.13 (moderate) and 0.26 (large). In the current study, since the R2 coefficients for the endogenous constructs entrepreneurial self-competency, entrepreneurial self-efficiency and student satisfaction are greater than the maximum threshold criterion of 0.26 postulated by Giao and Vuong, the structural model is deemed to be appropriate for the design of the study.

Having verified the adequacy of the measurement model, we proceeded to the structural model. In assessing the structural model, we tested the significance of the path coefficients. In addition to the predictive relevance of the model, the explanatory power of the structural model was assessed using the coefficient of determination (R2). The effect and significance of every path were confirmed by employing bootstrapped t-values (5000 subsamples), as suggested by Tortosa et al. [80]. The results of the final path analysis used to test the hypotheses are reported in Table 7. After performing a bootstrapping analysis in PLS-SEM for the structural model, entrepreneurial education was found to have a positive and significant influence on student satisfaction (b = 0.679, $p < 0.001$) and entrepreneurial self-efficacy (b = 0.302, $p < 0.001$). However, entrepreneurial education was found to have an indirect and insignificant relationship with entrepreneurial self-competency (b = −064, $p = 0.207$). Hypotheses $H_1$ and $H_2$ were therefore supported, but $H_3$ was not supported. Student satisfaction was also found to have a positive and significant influence on entrepreneurial self-efficacy (b = 0.274, $p = 0.002$) and entrepreneurial self-efficacy. Thus, hypotheses $H_4$ and $H_5$ were also supported. The final direct path (ESE–ESC) was also confirmed since entrepreneurial self-efficacy had a direct and significant influence on entrepreneurial self-competency (b = 0.656, $p < 0.001$). Hence, $H_6$ was also supported.

**Table 7.** Results of structural model assessment.

| Hypothesis | Path | Path Coefficient | $p$ Value | Result |
|---|---|---|---|---|
| $H_1$ | EE → SS | 0.679 | <0.001 | Supported |
| $H_2$ | EE → ESE | 0.302 | <0.001 | Supported |
| $H_3$ | EE → ESC | −0.064 | 0.207 | Not Supported |
| $H_4$ | SS → ESE | 0.274 | 0.002 | Supported |
| $H_5$ | SS → ESC | 0.141 | 0.041 | Supported |
| $H_6$ | ESE → ESC | 0.656 | <0.001 | Supported |
| $H_7$ | EE → SS → ESC | 0.096 | 0.044 | Supported |
| $H_8$ | EE → ESE → ESC | 0.198 | <0.001 | Supported |

4.2.2. Mediation Analysis

The study involved an analysis of the mediation effects of student satisfaction and entrepreneurial self-efficacy in the nexus between entrepreneurial education and entrepreneurial self-competencies. Within the literature, PLS-SEM reports two major types of mediation: full (complete) mediation and partial mediation. As reported by Hair et al. [81], there is full mediation when the direct effect is insignificant, but an indirect specific effect is significant. Partial mediation is said to exist when both direct and specific indirect effects are significant. Going by the above rule, the report in Table 8 points to full mediation roles for both student satisfaction and entrepreneurial self-efficacy.

**Table 8.** Specific indirect effects.

| | | | | | |
|---|---|---|---|---|---|
| EE → SS → ESC | 0.096 | 0.091 | 0.056 | 1.712 | 0.044 |
| EE → ESE → ESC | 0.198 | 0.196 | 0.055 | 3.588 | <0.001 |

This is because the indirect path (EE → SS → ESC) that shows the mediation role of student satisfaction in the link between entrepreneurial education and entrepreneurial self-competency is significant ($p = 0.044$) while the direct path (EE → ESC) is insignificant ($p = 0.207$; see Table 7) at a 0.05 significance level. Secondly, the indirect path (EE → ESE → ESC) that shows the mediation role of entrepreneurial self-efficacy in the link between entrepreneurial education and entrepreneurial self-competency is significant ($p = <0.001$) while the direct effect (EE → ESC) is insignificant ($p = 0.207$; see Table 7) at a 0.05 significance level.

## 5. Discussion

This study posited eight hypotheses and applied structural equation modelling with PLS to test the relationships. Seven of the hypotheses were supported, with one not supported.

First, the study found a direct relationship between entrepreneurial education and student satisfaction. This relationship is supported by similar findings in the literature [25,81]. The rationale behind this relationship is not unsubstantiated. The provision of quality entrepreneurial education is a service quality proposition; therefore, an improvement in service quality elicits student satisfaction with the programme. Therefore, if teaching quality, curriculum content, support services, apprenticeship opportunities, coaching and mentoring opportunities are embedded in the entrepreneurial programme, student satisfaction level increases. This finding is of theoretical significance as it has implications for the expectancy disconfirmation theory (EDT) described by Oliver [82], as well as for the social cognition theory (SCT) described by Badwan et al. [83]. The EDT posits that when users experience a lower service performance than was initially promised, they complain about the experience or refute it. However, when the offering meets or exceeds their expectation, they confirm the experience, which is reflected in a raised level of satisfaction. In the current study, the outcome suggests that the service offered by the HEIs exceeded respondents' expectations; hence, the respondents expressed satisfaction. Therefore, if HEIs offer quality entrepreneurial education, students' expectations are met, which further leads to enhanced student satisfaction with the programme [4]. Regarding the SCT, Bandura suggests that the environment can influence the individual's behaviour and vice versa. The current finding implies that a supportive educational environment can engender the desired entrepreneurial behaviours in students; said behaviours can also be applied by students to found start-ups to improve society. The unique relationship between entrepreneurial education and student satisfaction also has implications for the third United Nations sustainable development goal, which aims to provide quality education and lifelong learning for all. The outcome of this research suggests that quality entrepreneurial education can be defined from the viewpoint of student satisfaction. That is, entrepreneurial education is meaningful only if it meets students' quality expectations, which can then lead to the development of crucial entrepreneurial competencies and capabilities. HEIs in Ghana, as well as other developing and emerging economies, can make entrepreneurial education meaningful by providing satisfying learning experiences for their students.

Secondly, the study established a direct relationship between entrepreneurial education and students' entrepreneurial self-efficacy. A good number of studies in the literature lend credence to this finding, for example, Maritz and Brown [18], Sharma and Jamwal [32], Wardana et al. [33], Shahab et al. [35], Zhang et al. [38], Mahendra [84], Hoang et al. [85] and Setiawan and Lestari [86]. The findings, therefore, reveal that the institutions under study provided entrepreneurial knowledge and succeeded in helping the students develop self-belief and a can-do spirit with respect to starting their own projects. The students, thus, believed that they could start and progress their own social enterprises.

The third hypothesis tested the relationship between entrepreneurial education and entrepreneurial self-competencies. This hypothesis was not supported as the relationship

between the two constructs was found to be non-linear and insignificant. Contrary to expectations, entrepreneurial education did not have a direct influence on students' entrepreneurial competencies. This finding is unique and novel since this is the first study to establish that student satisfaction fully mediates the relationship between entrepreneurial education and entrepreneurial self-competency within the social enterprise sector. This finding emphasises the critical role that HEIs have to play in developing student satisfaction within their entrepreneurial learning experiences. The study found that entrepreneurial self-efficacy also fully mediates the relationship between entrepreneurial education and entrepreneurial self-competency. These relationships are explained by the fact that the respondents who expressed satisfaction and self-confidence associated such attributes with self-competency outcomes. The current finding negates the findings of Lv et al. [14] and Wei et al. [87], who posited that entrepreneurial education could directly cultivate entrepreneurial competencies among students. Nevertheless, that finding still finds support from the literature in the works of Bolzani and Luppi [13], Oosterbeek et al. [88] and Li et al. [89]. These authors found that the relationship between entrepreneurial education and entrepreneurial competence was indirect and mediated by intervening factors, such as entrepreneurship apprenticeships, entrepreneurship competitions, mentoring and coaching. The current finding is significant as it unravels the important roles of the mediating variables (entrepreneurial self-efficacy and student satisfaction), which are conspicuously missing in earlier studies that found a direct relationship between entrepreneurial education and entrepreneurial competencies.

This study also established a direct relationship between student satisfaction and entrepreneurial self-efficacy. An improvement in student satisfaction via quality entrepreneurial education leads to an improvement in entrepreneurial self-efficacy. This finding is supported by studies by Ahmed et al. [48], Yen and Lin [49], Wei, Liu, Sha [87] and Zeshan et al. [90]. The reason for this association is not farfetched. In the current study, students who were satisfied with the level of quality demonstrated in their entrepreneurship education, such as high levels of teacher expertise, interaction with successful entrepreneurs and opportunity for an apprenticeship, were more likely to develop a sense of belief in themselves and start their own social enterprises.

Furthermore, a positive association was established between student satisfaction and entrepreneurial self-competencies. This finding suggests that students who were satisfied with the quality of their entrepreneurship education developed the critical competencies required for them to start and nurture their own businesses. This finding is consistent with the theoretical finding of Elliot and Harackiewicz [91], who posit that student satisfaction can engender high levels of competency and skills performance. The findings of James and Cassidy [92] also lend support to the positive association between student satisfaction and competency development. The authors suggest that quality entrepreneurship education that incorporates authentic assessment can engender student satisfaction, which can further fuel entrepreneurial competencies.

In this study, we report a direct link between entrepreneurial self-efficacy and entrepreneurial self-competency. This position is supported by previous empirical findings [55,56,89,90]. Results from the study reveal that the majority of students who believed in their ability to accomplish certain tasks scored high on entrepreneurial self-competency. This association has strong support from the studies of Seikkula-Leino and Salomaa [55], who posit that students' sense of competence can be attributed to their confidence in their ability to attain set targets, overcome hurdles and reach for their dreams. Borba [93] posits that in any given community, members with high self-efficacy tend to develop effective communication skills and are likely to work towards achieving the goals of the group.

Finally, both student satisfaction and entrepreneurial self-efficacy were found to fully mediate the relationship between entrepreneurial education and entrepreneurial self-competencies. The implication is that entrepreneurial education cannot influence entrepreneurial self-competencies without student satisfaction and entrepreneurial self-efficacy. Although similar findings were reported in traditional entrepreneurial education

programmes by Ustyuzhina et al. [52] in a descriptive survey, these findings are quite novel and have significant policy implications for the design and implementation of social entrepreneurial education in higher education institutions. These findings also have implications for the incorporation of the eighth United Nations SDG into the curriculum of HEIs. According to Halsall et al. [94], the alignment of SDGs with the curriculum of HEI research programmes can enhance the pedagogic experience of students and have positive implications for the generation and sustainability of employment for students. With particular reference to the current study, the outcomes suggest that embedding the eighth SDG into Ghana's HEI entrepreneurship curriculum could lead to enhanced student satisfaction, self-efficacy and the development of entrepreneurial competencies. Most importantly, the development of the requisite self-belief and competencies may lead students to launch entrepreneurial initiatives and business start-ups.

## 6. Theoretical Implications

This study further established a direct relationship between entrepreneurial self-efficacy and entrepreneurial self-competency. This relationship has been corroborated in the extant literature on entrepreneurial education, for example, Bullock-Yowell et al. [53], Seikkula-Leino and Salomaa [55] and Bergman et al. [95] James and Casidy (2018) [92]; Borba, (1989) [93]. This particular finding from the current study is one of the first to confirm the ENTself (Assessment Framework of Entrepreneurial Competencies Integrating Self-esteem and Self-efficacy) theoretical framework of Seikkula-Leino and Salomaa [55]. The ENTself framework is an expansion of Bandura's [15] theory, which posits that self-efficacy is a major psychological influencer or function of competence. This positive association is explained by the fact that students who exhibit self-belief and have a heightened sense of achievement and purpose are psychologically well positioned to learn and develop the competencies required to start and develop new social enterprises. On the other hand, business students who do not possess self-efficacy and self-belief do not see the need to acquire the key competencies needed to commence and sustain social enterprises.

This study further establishes two unique theoretical contributions. The first contribution is that student satisfaction mediates the relationship between entrepreneurial education and entrepreneurial self-competency. This is the first study of its kind to establish student satisfaction as an intervening construct in the relationship between entrepreneurial education and entrepreneurial self-competencies within a social enterprise context. This finding was one of full mediation, in that entrepreneurial education could not engender entrepreneurial competencies without students being satisfied with the entire entrepreneurship programme mounted by the institution. This brings to the fore the crucial role of student satisfaction in building the desired competencies and skills in students for smoother transitioning into the world of entrepreneurship and career development. Secondly, regarding the second parallel mediation, the study found that entrepreneurial self-efficacy mediates the relationship between entrepreneurial education and entrepreneurial self-competency. This was also found to be a complete mediation, establishing the fact that self-efficacy is a crucial ingredient in building entrepreneurial competencies among students.

## 7. Managerial and Practical Applications

Managers, as well as academicians of social enterprise education in higher education institutions, should take due cognisance of the crucial role of student satisfaction in the implementation of entrepreneurial learning programmes. The findings in this study suggest that students can develop critical entrepreneurial competencies only when they are satisfied with a learning environment that provides students with a supportive environment. Additionally, practitioners of social enterprise education must ensure that students are exposed to the appropriate mix of business planning activities, entrepreneurial apprenticeship, extra-curricular activities, career guidance and counselling services, social

enterprise orientation, business trips and visits, well-equipped teaching rooms, guest entrepreneur speeches, and internships and work with entrepreneurs. These aspects should be embedded in the design and development of higher education curricula. Furthermore, findings from this study suggest that students' entrepreneurial self-efficacy can be developed through impactful coaching and mentoring activities together with the right mix of entrepreneurial pedagogy and heutagogy. By so doing, they avail themselves of opportunities to acquire the much-needed entrepreneurial competencies to start, grow and sustain their own social enterprise projects.

## 8. Limitations and Future Studies

The study was correlational by design and may therefore have issues of reliability. We, however, positioned ourselves to overcome this weakness by conducting a robust reliability analysis, as well as two-stage common method bias detection analyses. This notwithstanding, we recommend that longitudinal studies be conducted to enhance the reliability of the findings. In addition, further studies in other countries with different cultures and student attributes would be useful for improving the theoretical basis of this work, as we only focused on Ghanaian university students. The study may also suffer from a small sample size; however, we accounted for this weakness with a robust PLS-SEM analytical technique, as recommended in the literature [65]. This notwithstanding, studies with large sample sizes in other jurisdictions are recommended. The current study was also purely quantitative, and generalisations can only be limited to the Ghanaian context. The authors recommend that further studies adopt qualitative and mixed approaches to draw rich insights into the phenomena underlying the current study.

**Supplementary Materials:** The following supporting information can be downloaded at: https://www.mdpi.com/article/10.3390/su141912725/s1, File S1: Questionnaire for the study.

**Author Contributions:** Formal analysis, F.F.O. E.C.W.; Methodology, F.F.O., E.C.W., J.N.A.Q., E.O.A., E.C.O. and K.O.-A.; Supervision, M.S., F.F.O., E.C.W., D.H.-S., and J.P.H.; Writing – review & editing, F.F.O., E.C.W. and J.P.H. All authors have read and agreed to the published version of the manuscript.

**Funding:** This project was funded by the British Council and the APC paid by the University of Huddersfield.

**Institutional Review Board Statement:** Data collection was approved by the Ethics Review Committee of Accra Technical University.

**Informed Consent Statement:** Informed consent was obtained from all subjects in the study.

**Data Availability Statement:** Data will be made available upon request.

**Acknowledgements:** Financial support for the development of this article was received from the British Council as part of the Innovation for African Universities (IAU) project.

**Conflicts of Interest:** The authors declare no conflict of interest.

## Appendix A

**Table A1.** Harman's single-factor test for common method bias.

| Component | Initial Eigenvalues | | | Extraction Sums of Squared Loadings | | |
|:---:|:---:|:---:|:---:|:---:|:---:|:---:|
| | Total | % of Variance | Cumulative % | Total | % of Variance | Cumulative % |
| 1 | 12.964 | 44.703 | 44.703 | 12.964 | 44.703 | 44.703 |
| 2 | 4.470 | 15.415 | 60.118 | | | |
| 3 | 1.779 | 6.135 | 66.253 | | | |
| 4 | 1.453 | 5.010 | 71.263 | | | |
| 5 | 0.842 | 2.902 | 74.165 | | | |

| 6 | 0.762 | 2.629 | 76.794 |
| 7 | 0.668 | 2.303 | 79.097 |
| 8 | 0.536 | 1.848 | 80.945 |
| 9 | 0.490 | 1.689 | 82.634 |
| 10 | 0.439 | 1.514 | 84.148 |
| 11 | 0.382 | 1.317 | 85.465 |
| 12 | 0.375 | 1.293 | 86.758 |
| 13 | 0.371 | 1.279 | 88.036 |
| 14 | 0.361 | 1.244 | 89.280 |
| 15 | 0.324 | 1.119 | 90.399 |
| 16 | 0.301 | 1.038 | 91.436 |
| 17 | 0.294 | 1.014 | 92.451 |
| 18 | 0.272 | 0.937 | 93.388 |
| 19 | 0.266 | 0.918 | 94.306 |
| 20 | 0.239 | 0.824 | 95.130 |
| 21 | 0.236 | 0.813 | 95.943 |
| 22 | 0.212 | 0.730 | 96.673 |
| 23 | 0.192 | 0.661 | 97.334 |
| 24 | 0.157 | 0.540 | 97.874 |
| 25 | 0.143 | 0.493 | 98.367 |
| 26 | 0.132 | 0.456 | 98.823 |
| 27 | 0.122 | 0.420 | 99.243 |
| 28 | 0.118 | 0.407 | 99.650 |
| 29 | 0.102 | 0.350 | 100.000 |

Extraction method: principal component analysis.

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
