# Peer review of "The Nexus between Entrepreneurial Education and Entrepreneurial Self-Competencies: A Social Enterprise Perspective"

_sustainability, doi:10.3390/su141912725_

Round 1
Reviewer 1 Report
Dear Authors,
Thank you for the opportunity to read your paper entitled “The nexus between entrepreneurial education and entrepreneurial self-competencies: a social enterprise perspective”. Based on the entrepreneurial education literature and Social Learning theory, the authors formulated a conceptual model and tested several relationships. First, the (i) the influence of entrepreneurial education on student satisfaction, entrepreneurial self-efficacy, and entrepreneurial self-competencies. Second, the influence of student satisfaction on entrepreneurial self-efficacy, and entrepreneurial self-competencies. The third is the influence of entrepreneurial self-efficacy and entrepreneurial self-competency. Lastly, the mediation role of student education and entrepreneurial self-efficacy. My overall impression is that the paper is well written, and its structure is well developed as well. Moreover, it is based on a survey method conducted on 186 students from Ghana higher education institutions.
Besides these positive points, I have major concerns regarding this paper. Following this, I would like to point out each of my concerns to provide useful feedback for the authors.
First, I am not very convinced about the novelty of the paper as claimed by the authors. Most of the tested relationships have already been presented in previous studies and, mainly, what this paper does is to confirm (or not) previous studies. This is evident when you read the discussion section (e.g…. a good number of studies in the literature lend credence to this finding). In particular, the authors emphasize the mediation roles of student satisfaction and entrepreneurial self-efficacy. However, a further explanation of these findings is needed. Consequently, one important improvement would be to better position the paper´s contributions and advances regarding the other studies.
Second, the hypotheses are well developed. However, it is not clear which variables compose each investigated construct. Authors should provide an Appendix with the statements and individual variables for each construct. This aims to understand the nature of each construct.
Third, the research method is very solid but I would like to raise the following questions: (i) The section “sample and data collection” should be the first one in Section 3; (ii) What did you mean by the following sentence: “the study adopted a convenience sampling technique to obtain a targeted representative sample of 250 students”. Why 250 students? How were they selected? How do you justify the sample size? It seems too small considering the size of the population. (iii) Can you provide a profile of the respondents? What is the influence of the profile of the students on the results? I am saying this because senior students are in a better position to evaluate the quality of education, entrepreneurial skills, etc.; (iv) I would like to hear about Harman´s test for CMB.
Finally, my last comments address the results and contributions of the article. Interestingly, seven hypotheses were confirmed, which confirms the logical structure described by the conceptual framework. I also noted that authors assume a descriptive approach in writing the contributions sections. Some insights for improvements: (i) In general, they reaffirm the relationships described in the conceptual model. Thus, the authors should clarify the real contributions of the paper (what is being added? What is different from the current knowledge?). The three theoretical implications need further clarification to convince us about the novelty/contribution of the paper. (ii) Second, how to better position the paper for the Sustainability audience? See the number of Sustainability papers that were cited by our paper. I was wondering if the paper should be submitted to an Educational journal. In this sense, I invite the authors to discuss the results and findings considering the perspective of the Sustainable Goals. This appeared in the introduction only but it was discarded by the authors. This could be one interesting avenue for this article. (iii) Another major problem deals with the context in which the study was developed. What do we learn from the Ghana experience? In this way, it does not matter. Indeed, it could be any country since there are no contextual factors and boundaries discussed in the paper. In my opinion, the paper should benefit from a better contextualization.
Please find relevant publications in Sustainability and other similar journals, include them in your discussion of the state-of-the-art, reposition your work, and re-articulate its contribution.
Therefore, I am afraid that in its current form, the paper does not justify its publication. Sustainability is a top-ranked scientific journal in which the research design and contributions must be very well demonstrated. My apologies for my feedback. I hope you can improve our article based on my comments.
Author Response
Thank you for your helpful comments. The team has responded to each in turn and as suggested revised throughout the paper, including revsing some of the references and assertions. One table has now been included within the appendix. In particular, we have changed substantially (yellow highlight) :
Section 3 lines 307-330; lines 374-380 (added appendix table),
Section 5 lines 493-527, 530-533; 563-571
Many thanks

Reviewer 2 Report
The paper aims to examine the relationship between student satisfaction and entrepreneurial self-competencies in the higher education sector in Ghana. Using a cross sectional survey the study aimed at examining important research hypotheses. A structural equation model was used to capture robust estimates of relationships and to confirm existing theories. Important results are emerged and they are helpful.
However, the manuscript is too long and too many tables are used. My suggestion is to reduce the number of tables and use only important ones.
Table 1 is for VIF and no need of this table.
Table 3 is not needed.
P=.000 is used which should be p<.001
Author Response
The team is most grateful for your commentary which we have embraced and made the suggested amendments which have enhanced the paper.
Two tables have been removed with one added to the appendix. teh value has also been corrected.
Thankyou one again for your comments

Reviewer 3 Report
The authors deserve enormous credit for the ambitious task of researching a relatively unchartered topic of social analysis. It is exceptionally well structured and extremely well written. The conceptual scaffolding of key concepts are very well delineated and mapped out for the reader with clarity and utmost care. The literature review provides an excellent base to build their research analyses on. What I found particularly impressive was the relationship between student satisfaction and entrepreneurial education. The methodological design was robust and original. The research findings were well mapped out with real insights and authenticity. The conclusions are very clear and I would further suggest have important implications for the higher education sector in terms of the importance of embedding entrepreneurial education to student satisfaction as it impacts on on both student confidence and sustainable practice.
This is one of the best articles I have read in a long time which is utterly compelling. It will be widely cited globally given the rich conceptual, strong literature review and methodological framework and results. A tour de force.
Author Response
Thank you for your kind comments and helpful remarks. they are most valued. The team has made some changes (yellow highlights) to the paper and we are in gratitude for your support.
